Masticatory musculature of the African mole-rats (Rodentia: Bathyergidae)

http://orcid.org/0000-0001-9782-2358 Cox Philip G. 1 philip.cox@hyms.ac.uk
Faulkes Chris G. 2
Bennett Nigel C. 3
1 Department of Archaeology and Hull York Medical School, University of York , York , UK
2 School of Biological and Chemical Sciences, Queen Mary University of London , London , UK
3 Mammal Research Institute, Department of Zoology and Entomology, University of Pretoria , Pretoria , South Africa
Wilson Laura
Electronic publication date: 2020 Mar 24
Publication date: 2020
Volume: 8
Electronic Location ID: e8847
Received 2020 Jan 30; Accepted 2020 Mar 3
Copyright: © 2020 Cox et al.
Copyright year: 2020
Copyright holder: Cox et al.
License: This is an open access article distributed under the terms of the Creative Commons Attribution License, which permits unrestricted use, distribution, reproduction and adaptation in any medium and for any purpose provided that it is properly attributed. For attribution, the original author(s), title, publication source (PeerJ) and either DOI or URL of the article must be cited.
License URL: https://creativecommons.org/licenses/by/4.0/

Keywords: Masticatory muscles, DiceCT, Virtual reconstruction, Bathyergidae, Rodentia

Funding: DST-NRF SARChI Chair GUN 647560 The specimens were collected under a grant from the DST-NRF SARChI Chair (GUN 647560 to Nigel C. Bennett). The funders had no role in study design, data collection and analysis, decision to publish, or preparation of the manuscript.

==============================
The Bathyergidae, commonly known as blesmols or African mole-rats, is a family of rodents well-known for their subterranean lifestyle and tunnelling behaviour. Four of the five extant bathyergid genera (Cryptomys, Fukomys, Georychus and Heliophobius) are chisel-tooth diggers, that is they dig through soil with their enlarged incisors, whereas the remaining genus (Bathyergus) is a scratch-digger, only using its forelimbs for burrowing. Heterocephalus glaber, the naked mole-rat, is also a chisel-tooth digger and was until recently included within the Bathyergidae (as the most basally branching genus), but has now been placed by some researchers into its own family, the Heterocephalidae. Given the importance of the masticatory apparatus in habitat construction in this group, knowledge and understanding of the morphology and arrangement of the jaw-closing muscles in Bathyergidae is vital for future functional analyses. Here, we use diffusible iodine-based contrast-enhanced microCT to reveal and describe the muscles of mastication in representative specimens of each genus of bathyergid mole-rat and to compare them to the previously described musculature of the naked mole-rat. In all bathyergids, as in all rodents, the masseter muscle is the most dominant component of the masticatory musculature. However, the temporalis is also a relatively large muscle, a condition normally associated with sciuromorphous rodents. Unlike their hystricomorphous relatives, the bathyergids do not show an extension of the masseter through the infraorbital foramen on to the rostrum (other than a very slight protrusion in Cryptomys and Fukomys). Thus, morphologically, bathyergids are protrogomorphous, although this is thought to be secondarily derived rather than retained from ancestral rodents. Overall, the relative proportions of the jaw-closing muscles were found to be fairly consistent between genera except in Bathyergus, which was found to have an enlarged superficial masseter and relatively smaller pterygoid muscles. It is concluded that these differences may be a reflection of the behaviour of Bathyergus which, uniquely in the family, does not use its incisors for digging.

Introduction

The comparative anatomy of the masticatory, or jaw-closing, muscles in rodents has been a well-studied topic over many years (Wood, 1965; Turnbull, 1970; Woods, 1972; Woods & Howland, 1979; Woods & Hermanson, 1985; Ball & Roth, 1995; Thorington & Darrow, 1996; Druzinsky, 2010; Cox & Jeffery, 2011, 2015). Early classifications of rodents were based on masticatory muscle anatomy (Brandt, 1855), although modern phylogenies based on molecular data (Blanga-Kanfi et al., 2009; Fabre et al., 2012; Swanson, Oliveros & Esselstyn, 2019) have shown many of the myological similarities between taxa to be the result of convergent evolution rather than shared evolutionary history. Nonetheless, the highly specialised feeding system in rodents (including enlarged, ever-growing incisors and a lower jaw that can move antero-posteriorly with respect to the cranium) has ensured that the morphology of the jaw adductor muscles remains a relevant research topic in the field of functional morphology.

One group of rodents that is particularly interesting with regard to the jaw-closing muscles is the Bathyergidae, a family of subterranean rodents known as blesmols or African mole-rats. The family comprises at least 21 species (Van Daele et al., 2007; Faulkes et al., 2011, 2017; Burgin et al., 2018; Visser, Bennett & Jansen Van Vuuren, 2019) in five genera—Bathyergus, Cryptomys, Fukomys, Georychus and Heliophobius—all found in sub-Saharan Africa. A sixth monospecific genus, Heterocephalus (the naked mole-rat), was until recently also included within the Bathyergidae. However, it has been proposed (Patterson & Upham, 2014) that Heterocephalus glaber should be placed in its own family, the Heterocephalidae (Landry, 1957) based on the depth of the split from other bathyergids (c. 31.2 Ma) and a number of morphological characters. Nevertheless, it is important to note that from an evolutionary perspective the two families are still united as a monophyletic superfamily, the Bathyergoidea (Fig. 1). This division of mole-rats into two families is not universally supported (Visser, Bennett & Jansen Van Vuuren, 2019), but nonetheless has been reflected in a number of recent mammalian taxonomies (Wilson, Lacher & Mittermeier, 2016; Burgin et al., 2018). Accepting this classification, the Bathyergidae first diversified in the early Miocene, around 17.9 Ma (Patterson & Upham, 2014) with the earliest known fossils dating from this time as well (Mein & Pickford, 2008).

Figure 1 Genus-level phylogeny of the African mole-rats.

Tree based on mitochondrial 12S rRNA and cytochrome b sequence data and analysis of 3,999 nuclear genes (Faulkes et al., 2004; Ingram, Burda & Honeycutt, 2004; Davies et al., 2015). A chronologically calibrated scale in millions of years ago is illustrated beneath the tree, estimated using a molecular clock approach and using the bathyergid fossil Proheliophobius for calibration of genetic distances. Adapted from Faulkes & Bennett (2013).

African mole-rats are highly specialised for a fossorial lifestyle, spending much of their life underground in complex networks of burrows (Bennett & Faulkes, 2000). Five of the six genera (Cryptomys, Fukomys, Georychus, Heliophobius and Heterocephalus) are chisel-tooth diggers that dig tunnels with their incisors, whereas the remaining genus, Bathyergus, is a scratch digger, only using its limbs for digging (Stein, 2000). Chisel-tooth digging blesmols share a number of morphological adaptations to this behaviour, such as enlarged incisors, taller and wider skulls, enlarged temporal fossae and longer jaws (McIntosh & Cox, 2016a, 2016b; Samuels & Van Valkenburgh, 2009). These modifications have been shown to facilitate the production of high bite forces and wide gapes, both necessary for digging with the incisors (McIntosh & Cox, 2016a).

Despite the well-known osteological adaptations to digging seen in the bathyergid masticatory system, there are comparatively few in-depth studies of the jaw muscle anatomy of blesmols in the published literature. Tullberg (1899), in his large survey of rodent and lagomorph anatomy, illustrated some of the more superficial muscles of Georychus capensis, Cryptomys hottentotus and Bathyergus suillus. Boller (1970) and Van Daele, Herrel & Adriaens (2009) provide more detailed descriptions, but only of C. hottentotus and Fukomys species respectively. A study of subterranean rodents published by Morlok (1983) has a much broader coverage, including all bathyergid genera except Fukomys, which was split more recently from Cryptomys (Kock et al., 2006). However, the only detailed descriptions and figures of masticatory musculature in this work are of Cryptomys. Most recently, the masticatory musculature of H. glaber (now removed from the Bathyergidae as mentioned above) was described by Cox & Faulkes (2014) using digital dissection. That is the jaw-closing musculature was visualised and virtually reconstructed via diffusible iodine-based contrast-enhanced computed tomography (diceCT). This methodology, developed over the last decade (Metscher, 2009; Jeffery et al., 2011; Gignac & Kley, 2014; Gignac et al., 2016), uses iodine staining to increase the radio density of soft tissues and render them visible in CT scans. The technique is a useful complement to physical dissection, particularly when studying small specimens with complex, layered musculature.

The aim of this study is to describe the jaw-closing musculature of all five currently recognised genera of bathyergid mole-rats, in order to facilitate comparisons between them and also with the masticatory musculature of the closely related naked mole-rat. It is hypothesised that all chisel-tooth digging bathyergids have a similar arrangement of jaw adductor muscles, owing to the strong functional constraint of needing to produce a high bite force at a wide gape (McIntosh & Cox, 2016a). Furthermore, it is hypothesised that the relative proportions of the jaw adductors in chisel-tooth digging bathyergids are similar to those previously described in the naked mole-rat (Cox & Faulkes, 2014), owing to its similar mode of digging and its shared common ancestor with Bathyergidae. Finally, it is hypothesised that blesmols of the genus Bathyergus differ from other bathyergid genera in the relative proportions of their jaw-closing muscles, as these taxa are scratch diggers that do not use their incisors to construct burrows (Stein, 2000). In particular, the temporalis has been proposed to be particularly important in chisel-tooth digging (Samuels & Van Valkenburgh, 2009; McIntosh & Cox, 2016a), so this muscle is hypothesised to be relatively smaller in Bathyergus.

Materials and Methods

Sample and scanning

Five ethanol-preserved heads of bathyergid mole-rats were obtained from collections originating from the University of Pretoria. The work was approved by the animal ethics committee at the University of Pretoria AUCC 030110-002 and AUCC 040702-015. The specimens represented one species from each of the five currently recognised genera of Bathyergidae: B. suillus, C. hottentotus, Fukomys mechowi, G. capensis and Heliophobius argenteocinereus. As ethanol is known to reduce the efficacy of diceCT (Gignac et al., 2016), the concentration of the preserving fluid was gradually reduced from 70% ethanol down to distilled water over a period of 2 weeks. After another week in distilled water, the specimens were immersed in 4% phosphate-buffered formaldehyde solution. Finally, the specimens were transferred to a 7.5% solution of iodine-potassium iodide in formaldehyde for 3 months. The stained specimens were scanned using microCT at the Cambridge Biotomography Centre, University of Cambridge. The scans were performed at 164 kV and 165 μA (175 kV and 156 μA for B. suillus), with a 0.5 mm copper filter and a beryllium target. Voxels were isometric with dimensions between 0.026 and 0.046 mm. Further scanning details are given in Table S1. The diceCT stacks are archived and available from www.morphosource.org (the DOI of each stack is given in Table S1).

Digital reconstruction

Scans were imported as stacked TIFF files to Avizo 9.2 Lite (Thermo Fisher Scientific, Waltham, MA, USA) and the jaw adductor muscles of one side of the head were reconstructed. The side of the head chosen for reconstruction differed between specimens and was based on the quality of the staining and scanning in the left and right muscles. The muscles that were digitally reconstructed comprised the superficial masseter, deep masseter, zygomaticomandibularis, temporalis, medial pterygoid and lateral pterygoid, following the nomenclature of Cox & Jeffery (2011) and Cox & Faulkes (2014). Automatic thresholding of masticatory muscles was not possible owing to insufficient contrast difference between bone and muscle, and therefore the muscles were reconstructed using manual painting of selected slices and interpolation between them. Each muscle volume was subjected to a single application of the ‘smooth labels’ algorithm within the Avizo segmentation editor (size, 4; mode, 3D volume), and the volume of each muscle was recorded. Muscle masses were calculated from volumes assuming a muscle density value of 1.0564 g cm−3 (Murphy & Beardsley, 1974), although absolute mass values should be treated with caution as both iodine staining and formalin preservation are known to lead to soft tissue shrinkage (Vickerton, Jarvis & Jeffery, 2013). The percentage contribution of each muscle to total adductor muscle mass was also calculated for each specimen. In addition to the digital dissections, three of the specimens were also physically dissected (Bathyergus, Cryptomys, Georychus). Digital and physical dissections were compared both quantitatively (relative muscles masses) and qualitatively to ensure that muscle attachment sites and boundaries between muscle layers had been correctly identified in the diceCT scans. As congruence between the dissection methods was good (attachment sites correctly identified, relative muscle masses within 4%), digital dissection was deemed to be an accurate reflection of the morphology.

The reconstructed muscles were visualised by aligning them with a virtually reconstructed skull and mandible. Because scans of the unstained specimens were not available, and I2KI staining renders the reconstruction of bony material very difficult, an individual of the same species, but not the same specimen, was used to create each skull and mandible. A Bookstein warp (Bookstein, 1989) was then used to fit the bony elements to the reconstructed muscles.

Results

The percentage contribution of each muscle to total masticatory muscle mass are given in Table 1 (absolute masses are given in Table S2) and the percentage split between the masseteric complex, temporalis and pterygoid muscles for each specimen is shown in Fig. 2. Overall, it can be seen that the relative proportion of each jaw-closing muscle is largely consistent between Cryptomys, Fukomys, Georychus, Heliophobius and Heterocephalus, which all have a masseter forming 58–63%, a temporalis contributing 26–32%, and pterygoid muscles accounting for 8–11% of total muscle mass. B. suillus differs from the other blesmols somewhat, with a relatively larger masseter (69%) and relatively smaller pterygoids (5%). The morphology of each muscle is described below and shown in Figs. 3–6.

Table 1 Percentage of total masticatory muscle adductor mass occupied by each jaw-closing muscle in each mole-rat genus.

	Bathyergus	Cryptomys	Fukomys	Georychus	Heliophobius	Heterocephalus	
Superficial masseter	37.5	24.5	27.4	25.9	19.3	23.4	
Deep masseter	23.7	23.8	22.1	22.8	27.7	25.5	
Anterior ZM	2.7	1.8	2.2	2.7	5.3	2.9	
Posterior ZM	1.8	2.9	4.2	3.6	6.0	2.6	
IOZM	3.7	5.6	4.9	7.0	4.8	5.4	
Temporalis	26.0	31.8	26.2	28.0	26.1	32.2	
Medial pterygoid	3.7	7.7	9.7	7.8	7.2	6.1	
Lateral pterygoid	0.9	2.0	3.2	2.3	3.7	2.0	
Note:

Values for Heterocephalus glaber from Cox & Faulkes (2014). Absolute muscle masses given in Table S2.

Figure 2 Relative contributions of the masseter, temporalis and pterygoid muscles to total adductor muscle mass in each genus of Bathyergidae and Heterocephalidae.

Data for Heterocephalus from Cox & Faulkes (2014).

Figure 3 Masticatory muscles of Bathyergidae.

Left lateral view of a 3D reconstruction of the cranium, mandible and masticatory muscles of: (A) Bathyergus suillus; (B) Georychus capensis; (C) Cryptomys hottentotus; (D) Fukomys mechowi; (E) Heliophobius argenteocinereus. Abbreviations: azm, anterior zygomaticomandibularis; dm, deep masseter; iozm, infraorbital portion of the zygomaticomandibularis; sm, superficial masseter; t, temporalis. Scale bars = 5 mm.

Figure 4 Superficial master and pterygoid muscles of Cryptomys hottentotus.

Left lateral view of a 3D reconstruction of the cranium, mandible, superficial masseter and pterygoid muscles. Cranium and mandible transparent for visualisation of muscles attaching to medial mandibular surface. Abbreviations: lp, lateral pterygoid; mp, medial pterygoid; pr, pars reflexa of the superficial masseter; sm, superficial masseter. Scale bar = 5 mm.

Figure 5 Coronal diceCT slice of Bathyergus suillus.

MicroCT slice through the head of Bathyergus suillus stained with iodine potassium iodide. Abbreviations: azm, anterior zygomaticomandibularis (dark green); dm, deep masseter (dark blue); man, mandible; pr, pars reflexa of the superficial masseter (light blue); pzm, posterior zygomaticomandibularis (light green); sm, superficial masseter (light blue); t, temporalis (red); ten, tendon of temporalis. White line on 3D reconstruction shows position of slice. Scale bar = 5 mm.

Figure 6 Temporalis and zygomaticomandibularis muscles of Bathyergidae.

Left lateral view of a 3D reconstruction of the cranium, mandible, temporalis and zygomaticomandibularis of: (A) Bathyergus suillus; (B) Georychus capensis; (C) Cryptomys hottentotus; (D) Fukomys mechowi; (E) Heliophobius argenteocinereus. Abbreviations: azm, anterior zygomaticomandibularis; iozm, infraorbital portion of the zygomaticomandibularis; pzm, posterior zygomaticomandibularis; t, temporalis. Scale bars = 5 mm.

Superficial masseter

The superficial masseter is a large muscle in all blesmols, although only a small part of it can be seen in lateral view (Fig. 3). It represents about a quarter of the total masticatory musculature in most of the bathyergid genera. However, Heliophobius has a slightly reduced superficial masseter of about 19% total muscle mass, and Bathyergus has a greatly increased superficial masseter that occupies over 37% of the total musculature. This muscle has a tendinous origin via a small attachment site on the skull on the ventral surface of the zygomatic process of the maxilla. The tendon initially runs antero-medial to the deep masseter, but the muscle itself then wraps around the deep masseter to take a more lateral position. The superficial masseter then inserts along the ventral margin of the masseter all the way to the tip of the angular process. The muscle also wraps around the ventral mandibular margin and extends widely over the medial surface of the mandible, forming a pars reflexa (Figs. 4 and 5). This reflected component covers almost the entire medial angular process, leaving just a small area for the insertion of the medial pterygoid muscle.

Deep masseter

The deep masseter is also a large muscle in blesmols, contributing 22–28% of the total adductor muscle mass in all genera. It originates along the entire length of the ventro-lateral surface of the zygomatic arch, from the attachment site of the superficial masseter anteriorly, to the zygomatic process of the squamosal posteriorly. The insertion of the deep masseter is along the lateral mandibular surface just dorsal to the insertion of the superficial masseter. Thus, in lateral view, the deep masseter covers the posterior half of the mandible (Fig. 3). The separation between the superficial and deep masseter muscles was one of the most difficult aspects of the digital dissection, with these two muscles appearing continuous in some places (Fig. 5). However, the physical dissections of Bathyergus, Cryptomys and Georychus provided confidence that the muscles had been correctly reconstructed. No division of the deep masseter into anterior and posterior sections was identified.

Zygomaticomandibularis

The zygomaticomandibularis or ZM is a small to medium-sized component of the bathyergid masticatory system. It forms 10–13% of the total muscle mass in most genera, although this rises to 16% in Heliophobius and drops to 8% in Bathyergus. The ZM is divided into three sections—infraorbital, anterior and posterior—that were easily identifiable and separable in all specimens (Fig. 6). The anterior ZM originates from the medial surface of the zygomatic arch, with the attachment site spanning the posterior half of the jugal bone and the anterior part of the zygomatic process of the squamosal. The origin of the posterior ZM is immediately posterior to that of the anterior ZM and runs medially along the zygomatic arch until it meets the glenoid fossa. Both muscles insert in a fossa on the lateral surface of the mandible, with the anterior ZM having largely ventrally oriented fibres and the posterior ZM running somewhat anteriorly from origin to insertion. The anterior margin of the anterior ZM is at the level of the coronoid process of the mandible. Both muscles are covered by the deep masseter in lateral view.

The infraorbital portion of the ZM (IOZM) is usually the largest division of the ZM (although not in Heliophobius where it is smaller than the anterior ZM). It takes a wide origin across the anterior orbital wall and zygomatic process of the maxilla. The fibres then run ventrally and converge to a much narrower insertion area on the lower margin of the coronoid process, next to the attachment site of the anterior ZM. In most bathyergid genera, the IOZM origin is confined to the orbit, but in Cryptomys and Fukomys, a very small extension of the IOZM can be seen to push through the infraorbital foramen to take its origin on the rostrum (Figs. 6 and 7).

Figure 7 Transverse diceCT slice of Cryptomys hottentotus.

MicroCT slice through the head of Cryptomys hottentotus stained with iodine potassium iodide. Abbreviations: iozm, infraorbital portion of zygomaticomandibularis (dark green); on, optic nerve; t, temporalis (red). White line on 3D reconstruction shows position of slice. Scale bar = 5 mm.

Temporalis

The temporalis is large in all bathyergid genera, forming between 26% and 32% of the total muscle mass. It originates on the braincase, covering the parietal and the posterior part of the frontal bone (Fig. 3). The posterior limit of the temporalis on the skull is the nuchal crest, the medial border runs along the midsagittal line and anteriorly it extends into the orbit where it meets the posterior border of the IOZM (Fig. 6). Fibres from all across this wide origin converge on a small insertion on the anterior margin and medial surface of the coronoid process on the mandible. This gives the temporalis a fan-shaped morphology, with fibres from the orbital region running vertically and fibres from the nuchal crest running horizontally over the top of the zygomatic process of the squamosal. A tendon running through the middle of the muscle from the coronoid process upwards appears to divide the ventral part of the muscle into lateral and medial portions, inserting on the lateral and medial surfaces of the coronoid process respectively (Fig. 5). However, these portions come together in the dorsal part of the muscle and it is not possible to subdivide the temporalis here with any certainty. Thus, to avoid introducing errors, the temporalis has been reconstructed as a single component.

Medial pterygoid

The medial pterygoid comprises between 7% and 8% of the total masticatory muscle mass in Cryptomys, Georychus and Heliophobius. It is a little larger in Fukomys, representing almost 10% total muscle mass, but is notably smaller in Bathyergus occupying only 3% of the adductor musculature. The medial pterygoid has an elongated anterior portion that extends deeply into the pterygoid fossa where it takes its origin. There is also a smaller part of this muscle that originates on the lateral surface of the pterygoid flange that is ventral to the attachment of the lateral pterygoid muscle. From these attachment sites, the medial pterygoid runs ventro-laterally, fanning out somewhat, to insert on the medial surface of the angular process of the mandible, just dorsal to the ventral margin. The insertion site is elongate but narrow and bounded on all sides by the superficial masseter (Fig. 4).

Lateral pterygoid

The lateral pterygoid is a relatively small muscle forming 2–4% of the total adductor muscle mass, except in Bathyergus in which it is just under 1% of the total musculature. It takes its origin from the lateral surface of the pterygoid flange, just dorsal to a part of the medial pterygoid. From there, it runs laterally and posteriorly to an insertion site on the medial surface of the condylar process of the mandible.

Discussion

The technique of diceCT was successfully used to reveal the masticatory muscle anatomy of all five extant genera of blesmols. Despite being stored in ethanol for a number of years, which can reduce the contrast differences between soft tissues stained with I2KI (Gignac et al., 2016), the microCT images produced here were of good quality and allowed the different masticatory muscles to be distinguished from one another (Fig. 5).

The most notable finding from this study is consistency of the relative muscle proportions across the chisel-tooth digging bathyergid genera (Cryptomys, Fukomys, Georychus, Heliophobius). This supports our first hypothesis which predicted that the functional demands of needing to produce a high bite force at wide gape (McIntosh & Cox, 2016a) would lead to a constrained configuration of masticatory muscles across the family. In these genera, the masseter complex (including superficial and deep masseters, and all parts of the ZM) forms approximately 60% of adductor muscle mass, the temporalis represents around 30%, and the two pterygoid muscles together make up the final 10%. This distribution of muscle mass, with its dominant masseter, but also relatively large temporalis, has also been reported in a number of rodents, such as the mountain beaver, Aplodontia rufa, several members of the Sciuridae (Ball & Roth, 1995; Druzinsky, 2010), and the North American beaver, Castor canadensis (Cox & Baverstock, 2016). Notably, all of these rodents are sciuromorphous (Wood, 1965), that is they have an extension of the deep masseter on to the rostrum and they are all relatively distantly related to blesmols (Fabre et al., 2012). In contrast, more closely related rodents, from the suprafamilial clade Ctenohystrica to which blesmols belong, generally differ from the bathyergid pattern by having an even more dominant masseter (70% or more of total muscle mass) and a much reduced temporalis (15% or lower), for example Hydrochoerus (Müller, 1933), Hystrix (Turnbull, 1970) and Ctenomys (Becerra, Casinos & Vassallo, 2013). These rodents are hystricomorphous and have a substantial extension of the IOZM through the infraorbital foramen on to the rostrum. It is notable that the rodent species that more closely resemble bathyergids in the proportions of their jaw-closing muscles are those that require high bite forces at the incisors, either for processing mechanically demanding food items (Smith & Follmer, 1972) or for tree-felling (Rosell et al., 2005). It appears that the demands of chisel-tooth digging may have driven convergent evolution of a similar distribution of muscle mass in blesmols.

The second hypothesis of this study predicted that the chisel-tooth digging bathyergids would resemble H. glaber in their masticatory muscle anatomy. This hypothesis is also supported, with the relative proportions of each muscle in the naked mole-rat (Cox & Faulkes, 2014) being very similar to that seen in Cryptomys, Fukomys, Georychus and Heliophobius. Given this similarity, it is possible that this muscle arrangement is ancestral for the Bathyergoidea (the superfamily containing Heterocephalidae and Bathyergidae). However, given the strong pressures exerted on morphology by chisel-tooth digging (Lessa, 1990; Samuels & Van Valkenburgh, 2009; Gomes Rodrigues, Šumbera & Hautier, 2016; McIntosh & Cox, 2016a, 2016b), it is also possible that this configuration of adductor muscles evolved independently in the two families, especially given that they appear to have diverged over 30 million years ago (Patterson & Upham, 2014).

The exception to the common arrangement of masticatory muscles in the Bathyergidae is Bathyergus, the only genus of scratch-digging blesmols. The distribution of muscles in this genus is 69% masseter, 26% temporalis and 5% pterygoids. Thus, our third hypothesis that Bathyergus would differ from the chisel-tooth diggers is supported. However, it should be noted that only one specimen of each genus was available for study, so no statistical test of the difference between scratch and chisel-tooth diggers could be undertaken. We further predicted that the temporalis would be relatively smaller in the scratch digger, owing to the perceived importance of the temporalis in chisel-tooth digging (Samuels & Van Valkenburgh, 2009; McIntosh & Cox, 2016a), but this was not the case. The temporalis muscle in Bathyergus forms a similar proportion of total adductor muscle mass as in Heliophobius and Fukomys; instead the masseter complex in Bathyergus, in particular the superficial masseter, is relatively larger and the pterygoid muscles form a smaller part of the masticatory musculature. The lack of difference in the relative temporalis mass may reflect the fact that all bathyergid genera, Bathyergus included, have diets that incorporate hard foods such as the roots and tubers of geophytes, many of which are of large size and would require the use of a wide gape. Thus the size of the temporalis may be driven more by diet than by mode of digging.

The function of the superficial masseter has been debated by a number of authors, but it is generally thought to be important in the power stroke of both gnawing and chewing (Gorniak, 1977; Byrd, 1981) as well as being the main protractor of the lower jaw (Hiiemae, 1971), based on the antero-posterior orientation of the muscle fibres. Thus, the enlarged superficial masseter in Bathyergus may be an adaptation to its diet which incorporates many tough grasses and bulbs (Bennett & Faulkes, 2000). The function of the expansion of the superficial masseter on the medial mandibular surface, the pars reflexa, is less clear. Satoh & Iwaku (2004) have suggested that it may enable a wider gape by increasing the resting length of the muscle fibres. In most bathyergid genera, this would be advantageous as it would facilitate the wide opening of the jaws necessary for chisel-tooth digging. Blesmols of the genus Bathyergus do not dig with their teeth (Stein, 2000) but the males do fight with their incisors (Bennett & Faulkes, 2000), again requiring a wider gape. Whether or not the fighting behaviour requires a wider gape (and thus larger superficial masseter) than chisel-tooth digging is at present unclear.

The reduced pterygoid muscles in Bathyergus, particularly the medial pterygoid, may also be a reflection of scratch digging behaviour. It has previously been noted that the medial pterygoid contracts more strongly during incisor gnawing than molar chewing (Weijs & Dantuma, 1975). Thus, scratch digging mole-rats may not need such large pterygoid muscles as their chisel-tooth digging counterparts. Alternatively, the reduced medial pterygoid may simply reflect the reduced area for attachment on the medial surface of the angular process, resulting from the increased size of the pars reflexa of the superficial masseter in Bathyergus (Satoh & Iwaku, 2004).

In general, the muscle reconstructions presented here are in agreement with the previous descriptions of bathyergid jaw musculature given by Tullberg (1899) and Morlok (1983), but differ in some respects from the anatomy reported by Boller (1970) and Van Daele, Herrel & Adriaens (2009). The main difference arises in the morphology of the superficial masseter, which in Boller (1970) and Van Daele, Herrel & Adriaens (2009) is reported to have a wide expansion across the deep masseter in lateral view and is split into sections known as M1a, M1b and M2. Here, we agree with Morlok (1983) that the main part of the superficial masseter in lateral view is very slender and runs along the ventral margin of the mandible and that most of the muscle attaching to the lateral surface of the mandible is the deep masseter. It is clear that the division between the superficial and deep masseter muscles is quite difficult to determine in some places, but following both digital and physical dissection, we are confident that the morphology presented here is correct and moreover resembles that reported for the naked mole-rat (Cox & Faulkes, 2014).

The other major difference between the reconstructions here and that of Boller (1970) is with regard to the temporalis. The illustrations of Cryptomys in Boller (1970) show the temporalis extending ventrally on to the mandible, between two sections of the ZM muscle. We believe this ‘pars zygomatica of the temporalis’ to be a misidentification of the anterior ZM resulting from the close apposition of the two muscles, and the insertion of the temporalis on the mandible to be restricted to the coronoid process.

The virtual reconstructions presented here highlight one well-known peculiarity of bathyergid musculature—the lack of jaw-closing muscles attaching to the rostrum in this family. Species in the Ctenohystrica, which includes the blesmols, are almost all hystricomorphous; that is they have a greatly enlarged infraorbital foramen, through which a portion of the ZM muscle (the IOZM) extends to take its origin on the rostrum (Hautier, Cox & Lebrun, 2015). In the Bathyergidae, the infraorbital foramen is much smaller and very little, if any, muscle passes through it. This is very similar to the condition known as protrogomorphy, which is believed to be the ancestral state for rodents (Wood, 1965). Here we have designated the rostral most section of the ZM as the ‘IOZM’, but only in Cryptomys and Fukomys does it pass through the infraorbital foramen. In the remaining three genera, the IOZM is confined to the orbit. Maier & Schrenk (1987) noted that some muscle fibres pass through the infraorbital foramen in Bathyergus and Georychus in early ontogeny, but subsequently retreat and are absent from the rostrum at birth. Similarly, no part of the IOZM was found on the rostrum in the specimens of these two genera in this study.

Despite not extending on to the rostrum, the IOZM is usually the largest part of the ZM in bathyergids and has a wide origin across the anterior part of the orbit. Indeed, in all the specimens studied here, its posterior margin meets the anterior margin of the temporalis in the orbit. Such an arrangement of muscles has likely been made possible by the extreme reduction of the eye in these fossorial species, which has left space into which the muscles have expanded (Fig. 7). The large IOZM also gives a clue to the evolutionary history of the masticatory muscles in Bathyergidae. Although frequently referred to as being ‘protrogomorphous’ (Tullberg, 1899; Wood, 1965, 1985), the morphology of the bathyergid ZM muscle does not resemble that of the extant protrogomorph, Aplodontia rufa, in which the ZM origin is restricted to the zygomatic arch and does not extend dorsally into the orbit. Instead, the bathyergid IOZM more closely resembles that of other hystricomorphs, minus the extension on to the rostrum, a morphology that seems more likely to be secondarily derived than ancestrally retained. This hypothesis is also supported by the phylogenetic position of bathyergids within the otherwise hystricomorph clade Ctenohystrica (Swanson, Oliveros & Esselstyn, 2019), the presence of hystricomorphy in some fossil bathyergids (Lavocat, 1973), and the previously mentioned presence of hystricomorphy in early development of some blesmols (Maier & Schrenk, 1987).

The loss of the rostral extension of the IOZM seems an unusual morphological change, given that this muscle is known to improve the efficiency of molar chewing in rodents (Cox et al., 2012; Cox, 2017). It is possible that it is an adaptation towards increased use of the incisors in digging, as has been suggested for H. glaber (Cox & Faulkes, 2014). Shortening the rostrum would decrease the out-lever of incisor biting and would thus increase bite force, but would leave less room for rostral muscle attachment. In addition, the loss of the IOZM from the rostrum could be a strategy for increasing maximum gape, which is also important in chisel-tooth digging (McIntosh & Cox, 2016a). It should be noted that Bathyergus also lacks the rostral portion of the IOZM, despite being a scratch digger. This may be a case of phylogenetic inertia and that having lost the rostral IOZM once in its evolutionary history, Bathyergus has not re-evolved it.

Conclusion

The masticatory musculature of the Bathyergidae is dominated by the masseter muscle, but also has a relatively large temporalis, similar to the condition seen in many sciuromorph rodents. The ZM muscle does not extend on to the rostrum (except very slightly in Cryptomys and Fukomys), a condition that is thought to be secondarily derived from a hystricomorph ancestor. The relative proportions of the jaw-closing muscles are largely consistent between the chisel-tooth digging blesmols, but the scratch digging genus, Bathyergus, differs in having a larger superficial masseter and smaller pterygoid muscles. Despite the deep split between the Heterocephalidae and the Bathyergidae, the jaw adductor musculature of the naked mole-rat is very similar to that of the chisel-tooth digging bathyergids.

Supplemental Information

Supplemental Information 1 Specimen information and scanning parameters.

Specimen IDs, DOIs and microCT scanning parameters for each specimen used in this analysis.

Click here for additional data file.

Supplemental Information 2 Masticatory muscle masses.

Absolute masses (in g) of masticatory muscles of African mole-rats. Data for Heterocephalus from Cox & Faulkes (2014).

Click here for additional data file.

The authors thank Keturah Smithson (Cambridge Biotomography Centre, University of Cambridge) for microCT scanning the specimens.

Additional Information and Declarations

Competing Interests

Author Contributions

Animal Ethics

Data Availability

Philip G. Cox is an Academic Editor for PeerJ.

Philip G. Cox conceived and designed the experiments, performed the experiments, analysed the data, prepared figures and/or tables, authored or reviewed drafts of the paper, and approved the final draft.

Chris G. Faulkes conceived and designed the experiments, analysed the data, prepared figures and/or tables, authored or reviewed drafts of the paper, and approved the final draft.

Nigel C. Bennett conceived and designed the experiments, analysed the data, authored or reviewed drafts of the paper, and approved the final draft.

The following information was supplied relating to ethical approvals (i.e., approving body and any reference numbers):

The work was approved by the animal ethics committee at the University of Pretoria (AUCC 030110-002 and AUCC 040702-015).

The following information was supplied regarding data availability:

All specimens are stored in the PalaeoHub, Department of Archaeology, University of York.

MicroCT stacks are available at Morphosource: M54784-98703, M54785-98704, M54786-98705, M54787-98706, M54788-98707.

The full details of specimens, scanning parameters and microCT stack files are available in Table S1.

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
