# Peer review of "Masticatory musculature of the African mole-rats (Rodentia: Bathyergidae)"

_PeerJ, doi:10.7717/peerj.8847_

## Round 0.1 · original submission · Minor Revisions

Both reviewers have commented positively that this study represents a welcome addition to the literature on cranial musculature in rodents and that it will be suitable for publication in PeerJ following some minor revisions. I agree with these comments and commend the authors on a well-written and carefully executed study. Besides the helpful suggestions of the reviewers, upon reading I have a few additional points:

1. Please provide the raw data (rather than just the percentages) for muscle masses
2. Please add in some further details concerning how the comparisons between the digital dissection and the manual dissection were conducted - this is mentioned on ln177, however I missed some information on how accuracy was determined - how congruent were the volumes/masses for these two data sets?
3. Please provide a few more sentences about how the smoothing was controlled across specimens (ln172) and/or what settings were used for this. The digital volume data will be directly impacted by these procedures.

·

Basic reporting

no comment

Experimental design

no comment

Validity of the findings

no comment

Additional comments

I have now read the paper by Cox and co-authors on the masticatory musculature of the African mole-rats (Rodentia: Bathyergidae) submitted to PeerJ. I found the paper interesting, the anatomical descriptions solid and the interpretation of the data meaningful and accurate. As such I mostly have minor suggestions to offer.

One thing I did find disappointing is that the authors did not report any data on muscle fibre lengths and pennation angles. This would make the manuscript much more useful for future studies interested in comparative functional anatomy or interested in creating biomechanical models which is one of the goals of the study. This is may only major comment on an otherwise excellent paper.

line 138 and elsewhere: the focus in the paper is on tooth digging which is fine. However, these animals are known to eat hard foods like roots and tubers which often are also of large size. As such having a large temporalis may also be of use in this context and may explain why the authors found no difference between the scratch digger and the tooth diggers. I would like to see this discussed/highlighted more in the manuscript.

line 326: I would like to have seen just a short comment on the fact that the authors had only a single specimen of each species in their data set. This prevents the authors from doing any statistical analyses testing for differences between tooth and scratch diggers and as such caution should be used in interpreting the differences.

·

Basic reporting

No comment

Experimental design

No comment

Validity of the findings

No comment

Additional comments

This paper presents a detailed comparative study of the jaw adductor musculature of five extant African mole-rats (bathyergids), and the naked mole-rat, using diceCT. The relative size and attachments of the muscles were discussed in relation to chisel-tooth digging (4 of the 5 bathyergids plus the naked mole-rat) versus scratch-digging (1 species of bathyergid).
Overall, the paper is very well written, the descriptions of the muscles are very succinct, the figures are informative, and the discussion of the results is comprehensive. It is also good to see the scans are available on Morphosource. The discussion is hypothesis driven and clear, and the conclusions are supported. I particularly like the discussion of convergent forms between chisel-tooth digging mole-rats and other rodents like beavers that also use their incisors for demanding tree-felling, and the function of the rostral portion of the IOZM (or lack thereof) in bathyergids.
I really don’t have much to add for improvements; I found two minor grammar changes I would make, but that is more a preference (on Line 101 I’d remove “is they” and only have “…chisel-tooth diggers, that dig tunnels…”; and, on Line 232-233, I’d have “posterior to” instead of “posterior that of”).
In the methods, please also add which specimens were physically dissected on Line 178-179, and refer to these in the Results at Line 222 as well. If not all species were physically dissected, an explanation for this should also be given.
Otherwise, I’m happy to recommend this for publication.

---

## Round 0.2 · accepted · Accept

Thank you for addressing the very minor comments raised by the two reviewers and my own reading of the manuscript. The addition of a supplementary data table, along with minor additions to the text, address in full the suggestions made by the reviewers. I'm happy to recommend your manuscript for publication and look forward to seeing it in press.